

# Nitric oxide (NO) in the Bohai and Yellow Seas

Ye Tian[1,3], Chao Xue[1,3], Chun-Ying Liu[1,3,*], Gui-Peng Yang[1,2,**], Pei-Feng Li[1,3], Wei-Hua Feng[4], and Hermann W. Bange[5]

[1]Key Laboratory of Marine Chemistry Theory and Technology, Ministry of Education, Qingdao, 266100, China
[2]Laboratory for Marine Ecology and Environmental Science, Qingdao National Laboratory for Marine Science and Technology, Qingdao 266071, China
[3]College of Chemistry and Chemical Engineering, Ocean University of China, Qingdao, 266100, China
[4]Key Laboratory of Engineering Oceanography, Second Institute of Oceanography, SOA, Hangzhou, 310012, China
[5]GEOMAR Helmholtz-Zentrum für Ozeanforschung Kiel, Kiel, 24105, Germany

*Correspondence to*: *Chun-Ying Liu (roseliu@ouc.edu.cn); **Gui-Peng Yang (gpyang@ouc.edu.cn)

**Abstract.** Nitric oxide (NO) is a short-lived compound of the marine nitrogen cycle; however, our knowledge about its oceanic distribution and pathways is rudimentary. Here we present the measurements of dissolved NO in the surface and bottom layers at 75 stations in the Bohai Sea (BS) and Yellow Sea (YS) in June 2011. Moreover, NO photoproduction rates were determined at 27 stations in both seas. The NO concentrations in the surface and bottom layers were highly variable and ranged from below the detection limit (i.e. 32 pmol L$^{-1}$) to 616 pmol L$^{-1}$ in the surface layer and to 482 pmol L$^{-1}$ in the bottom layer. There was no significant difference between the mean NO concentrations in the surface (186 $\pm$ 108 pmol L$^{-1}$) and bottom (174 $\pm$ 123 pmol L$^{-1}$) layers. A decreasing trend of NO bottom layer concentrations salinity indicates a NO input by submarine groundwater discharge. NO in the surface layer was supersaturated at all stations during both day and night and therefore the BS and YS were a persistent source of NO to the atmosphere at the time of our measurements. The accumulation of NO during daytime was resulting from photochemical production and photoprodcution rates were correlated to illuminance. The persistent nighttime NO supersaturation pointed to a, so far unknown, NO dark production. NO sea-to-air flux densities were much lower than the NO photoproduction rates. Therefore, we conclude that the bulk of the NO produced in the mixed layer was rapidly consumed before its release to the atmosphere. Overall, the oceanic NO emissions to the atmosphere were negligible compared to anthropogenic NO$_x$ sources such as emissions from ships.

**Introduction**

Nitric oxide (NO) is a short-lived intermediate of the oceanic nitrogen cycle (Bange, 2008). It received limited attention so far because its determination in seawater is challenging (Zafiriou and McFarland, 1980; Lutterbeck and Bange, 2015; Liu et al., 2017). NO in surface seawater can be produced via the photolysis of nitrite (NO$_2^-$) (Zafiriou and McFarland, 1981; Olasehinde et al., 2009; 2010; Liu et al., 2017):

$$NO_2^- + H_2O \xrightarrow{h\nu} NO + OH^-.$$



This reaction may account for 10 % of nitrite loss in surface waters of the central equatorial Pacific (Zafiriou et al., 1980). Zafiriou and McFarland (1981) suggested that photochemically produced NO is a potential source of atmospheric NO during daylight. Apart from the photochemical production, various microbial pathways of NO have been identified including denitrification, nitrification and anammox (Schreiber et al., 2014; Martens-Habbena et al., 2015; Caranto and Lancaster,
2017; Kuypers et al., 2018). Additionally, NO is a messenger molecule in marine organisms: phytoplankton does not only response to exogenous NO (Zhang et al., 2005), but also produce NO during their growth (Zhang et al., 2006a, b; Kim et al., 2006, 2008). Chen et al. (2015) reported that calmodulin (a messenger protein expressed in eukaryotic cells) of the tropical sea cucumber participates in the production of NO during immune response. Morrall et al. (1998; 2000) characterized the NO synthase activity in the tropical sea anemone *Aiptasia pallida*, suggesting that NO and NO synthase can act as
ecotoxicological biomarkers in the tropical marine environment. Moreover, the characterization of NO synthase gene in the intertidal copepod *Tigriopus japonicas* has been found and the intracellular production of NO in shrimp haemocytes has been observed (Xian et al., 2013; Jeong et al., 2016). Thus, NO seems to be widespread with different functions in marine organisms.

Current understanding of the oceanic NO distribution is mainly limited to the ocean surface (Zafiriou and McFarland, 1981;
Olasehinde at al., 2009; 2010; Liu et al., 2017) and oxygen minimum zones (Ward and Zafiriou, 1988; Lutterbeck et al., 2018). Only recently, the distribution of NO as well as its seasonal variation in the Jiaozhou Bay and adjacent waters were studied (Feng et al., 2011; Xue et al., 2012; Tian et al., 2016).

In this study, we present first measurements of dissolved NO in the Yellow Sea (YS) and the Bohai Sea (BS). The overarching objective of our study was to decipher the biogeochemical fluxes of NO in the BS and YS. The specific
objectives were (i) to determine the spatial variation of dissolved NO concentrations in the water column, (ii) to determine NO photoproduction rates, and (iii) to estimate the sea-to-air gas exchange fluxes of NO.

## 2 Materials and methods

### 2.1 Study area

The BS and the YS are marginal seas of the western Pacific Ocean. The BS is a shallow and almost enclosed sea with a
surface area of $77 \times 10^3$ km$^2$ and a volume of $1.39 \times 10^3$ km$^3$. Its maximum depth is 83 m with an average depth of only 18 m. The Huanghe River is a major source of freshwater to the BS whereas the YS is the source of salt for the BS through water exchange via the Bohai Strait. The YS has a surface area of $380 \times 10^3$ km$^2$ and total volume of $16.7 \times 10^3$ km$^3$. Maximum depth is 140 m with an average depth of 44 m, and like the BS, it is a continental shelf sea. The Yangtze River at the southwest corner of the YS is the major source of freshwater for the southern YS and the East China Sea. The hydrographic
properties of this region are mainly influenced by the Yellow Sea Cold Water Mass (YSCWM) on the shelf (Lü et al., 2010; Li et al., 2016) and the Yellow Sea Coastal Currents on the western side of the basin (Su, 1998; Lee et al., 2002; Zhang et al.,





2004). The BS and YS are regions surrounded by areas of high population growth and economic development in China and Korea (Zhan et al., 2010; Jiang et al., 2014).

## 2.2 Sampling

Samples were collected from 13 to 28 June 2011 on board of the R/V "Dong Fang Hong 2" in the BS and the YS. Sampling for NO from both the surface (at 1 m) and the bottom layer (= 1 m above the ocean bottom) were performed at 75 sampling stations (including one 24 h anchor station: B65) shown in Fig. 1 and listed in Table 1. Water samples were collected using 8-liter Niskin bottles equipped with silicon O-rings and Teflon-coated springs and mounted on a Sea-Bird CTD (conductivity, temperature, depth) instrument (Sea-Bird Electronics, Inc., USA). A 500 mL Wheaton glass serum bottle was rinsed with in situ seawater three times, and then was filled with seawater quickly through a siphon. When the overflowed sample reached the half volume of the bottle, the siphon was withdrawn rapidly, 0.5 mL saturated $HgCl_2$ (aq) solution was added, and the bottle was sealed quickly. The surface water samples were immediately analyzed after collection, and samples from other depths were temporarily placed in dark in a water bath with a surface seawater circulation system and analyzed within 1 hour (Liu et al., 2017). Photoproduction rates were determined in surface water samples from 22 stations in the YS (H01, H05, H08, H10, H17, H21, H25, H29, H31, H35, H37, H39, H42 B01, B05, B10, B12, B15, B18, B23, B27, and B35) and 4 stations in the BS (B42, B47, B51, and B68), respectively.

## 2.3 Analytical procedures

Analysis for dissolved NO was conducted with the improved method of Liu et al. (2017) by a combined purge-and-trap and fluorometric detection method. The precision of the analytical method was better than $\pm 7$ % and the limit of detection (LOD) was 32 pmol $L^{-1}$ (Liu et al., 2017).

The photolysis experiments were conducted under natural light conditions on deck. Surface water samples were filtered with 0.45 μm Millipore membrane and transferred into 10 mL cleaned quartz vials wrapped with Al foil with no headspace. Then 200 μL $NaN_3$ solutions were added. After the addition of 10 μL $1 \times 10^{-3}$ mol $L^{-1}$ 2, 3-diaminonaphthalene solution and gentle mixing, the fluorescence of the mixed solution was measured before irradiation (Liu et al., 2017). Then capped with a Teflon-lined silicone septum and without Al foil, the vial was placed in a shallow circulating seawater bath. The fluorescence of the solution was analyzed after irradiation for 0.5 hour. For dark controls, vials were wrapped in Al-foil. The quartz vials and syringe used in the experiment were soaked in a 10 % (v/v) HCl bath for 24 h, rinsed with Milli-Q water and baked at 500 ℃ for 4 hours.

Chlorophyll a (Chl-a) was fluorometrically measured by an F-4500 fluorescence spectrophotometer after filtration of 200 mL seawater through a Whatman glass fiber filter and extraction in 90 % acetone according to Strickland and Parsons (1968). The wind speeds were measured at a height of 10 m above the sea surface using a Model 27600-4X ship-borne weather instrument (Young, USA). Illuminance was measured by a digital illuminometer (TES-1330A, Shenzhen, China). Dissolved oxygen (DO) was measured in discrete water samples by the Winkler method (Grasshoff et al., 2009). The concentrations of




dissolved inorganic nitrogen (nitrate, nitrite, and ammonium) were analyzed using a nutrient automatic analyzer (Auto Analyzer 3, SEAL Analytical, USA) in the laboratory. The detection limits were 0.14 $\mu$mol $L^{-1}$ for nitrate, nitrite, and ammonium, with the precision of the method exceeding 3 % (Liu et al., 2005).

**2.4 Calculation of NO flux and NO saturation**

Fluxes of NO across the sea-to-air interface were estimated following the approach of McGillis et al. (2000) for a sparingly soluble gas which is also moderately reactive in the atmosphere:

$F = k_{sea} (c_{sea} - pNO_{air} \times H^{cp})$,

here $F$ stands for the flux density (mass area$^{-1}$ time$^{-1}$) across the air-sea interface, $k_{sea}$ is the gas transfer velocity (length time$^{-1}$), and $c_{sea}$ is the measured concentration of NO in the surface seawater (mass volumn$^{-1}$). The partial pressure of the

atmospheric NO ($pNO_{air}$) was calculated as:

$pNO_{air} = x'NO_{air} \times (p_{ss} - p_w)$,

where $x'NO_{air}$ is the mole fraction of atmosphere NO (dimensionless). We used the value of 2.13 ppb for $x'NO_{air}$ which is the average atmospheric NO mole fraction over the YS (Hu Min, Peking University, personal communication, 2018). And $p_{ss}$ is the barometric pressure at sea surface which was set to 1 atm as the average pressure and $p_w$ is the water vapor pressure

at sea surface which was calculated after Weiss and Price (1980):

$\ln p_w = 24.4543 - 6745.09 / (T + 273.15) - 4.8489 \times \ln (T + 273.15) / 100) - 0.000544 \times S)$.

$H^{cp}$ is the Henry's law constant which is calculated as:

$H^{cp}(T) = H^{\Theta} \times \exp (- \Delta sol H / R \times ( 1 / T - 1 / T^{\Theta} )$

where $-\Delta sol \frac{H}{R} = \frac{dlnH}{dln(\frac{1}{T})}$.

$H^{\Theta}$, and $-\Delta sol H / R$ are tabulated in Sander (2015).

$k_{sea}$ was calculated as:

$k_{sea} = k_w (1 - \gamma_a)$,

$\gamma_a = 1 / (1 + (k_a / (H^{cc} \times k_w))$,

$H^{cc} = H^{cp} \times RT$,

$k_a = 659 \times u \times (M_{NO} / M_{H_2O})^{-1/2}$,

$k_w = 0.251 \times u^2 \times (Sc / 660)^{-1/2}$,

where $k_w$ is the water side air-sea gas transfer coefficient for sparingly soluble gases (length time$^{-1}$) calculated according to Wanninkhof (2014), $\gamma_a$ is the fraction of the entire gas concentration gradient across the airside boundary layer as a fraction of the entire gradient from the bulk water to the bulk air (dimensionless) (McGillis et al., 2000), $k_a$ is the air side air-sea gas

transfer coefficient (length time$^{-1}$) according to McGillis et al. (2000), $H^{cc}$ is the Henry coefficient (dimensionless) (Sander, 2015 ), $M_{NO}$ and $M_{H_2O}$ are relative molecular mass of NO and $H_2O$ (dimensionless), and $u$ is the wind speed at 10 m height under neutral boundary conditions (length time$^{-1}$).





The Schmidt number (Sc) is the kinematic viscosity of water divided by the molecular diffusion coefficient of the gas in (sea)water (Jähne et al., 1987; Wanninkhof, 2014). Seawater dynamic viscosity ($\mu_{sw}$) is a function of temperature ($T$) and salinity ($S$) and was estimated using the following equations (Sharqawy et al., 2010):

$$\mu_{sw} = \mu_w (1 + A\,S + B\,S^2),$$

$$A = 1.541 + 1.998 \times 10^{-2}\,T - 9.52 \times 10^{-5}\,T^2,$$

$$B = 7.974 - 7.561 \times 10^{-2}\,T + 4.724 \times 10^{-4}\,T^2,$$

$$\mu_w = 4.2844 \times 10^{-5} + (0.157\,(T + 64.993)^2 - 91.296)^{-1}.$$

Seawater density was estimated using Millero's empirical equation (Millero et al., 1976), and NO diffusion coefficient in water was calculated according to Wise and Houghton (1968):

$$D_L = 0.9419\,\exp(0.0447\,T).$$

The saturation factor ($\alpha$) is defined as $\alpha = c_{sea} / (pNO_{air} \times H^{cp})$, $\alpha > 1$ represents NO was supersaturated and the flux was from sea to air.

## 3 Results and Discussion

### 3.1 NO in the surface and bottom layers

The NO concentrations from the surface and bottom layers of the BS and the YS as well as the local sampling time, bottom depth (D), temperature (T), salinity (S), Chl-$a$, wind speed (u), and DO are listed in Table 1. The surface concentrations of NO ranged from below the LOD to 616 pmol $L^{-1}$ with an overall average of 186 $\pm$ 108 pmol $L^{-1}$ and exhibited a considerable spatial variability (Fig. 2a). The mean NO surface concentrations in the BS (203 $\pm$ 107) and northern YS (NYS) (212 $\pm$ 130) were higher than the mean NO concentration in the southern YS (SYS) (159 $\pm$ 84). The NO concentrations in the bottom

layer ranged from below the LOD to 482 pmol $L^{-1}$, with an overall average of 174 $\pm$ 123 pmol $L^{-1}$. The mean concentrations of NO in the bottom layers of the BS and NYS were 228 $\pm$ 116, 210 $\pm$ 138, respectively and were higher than the mean (127 $\pm$ 98 pmol $L^{-1}$) for the SYS. The maximum NO surface and bottom concentrations were measured at stations B21 and B28 in the NYS, respectively (Fig. 2b), whereas the lowest NO surface and bottom concentrations were measured in the center of the SYS (Table 1). Overall, there were no statistically significant differences between the mean NO concentrations in the

surface and bottom layers.

NO surface concentrations did not show any statistically significant relationship with depth, sea surface temperature, salinity, DO, Chl-$a$, and illuminance. Trends of NO concentrations with salinity and DIN were was only found for the stations affected by the outflow of the Huanghe River in the southern BS (Fig. 3) where we found an inverse relationship between salinity and NO surface concentrations. High NO concentrations were associated with high DIN concentrations (data not

shown, see Liu et al., 2015; Yang et al, 2015) indicating that DIN, especially $NO_2^-$, was a prerequisite for enhanced NO concentrations.



The water columns of the BS and YS were well-oxygenated during our study and, thus, no suboxic or anoxic conditions were detected. Therefore, we did not found any enhancement of NO concentrations in the bottom layers with low DO or $NO_2^-$ concentrations as observed in the OMZ of the eastern tropical South Pacific Ocean off Peru (Lutterbeck et al., 2018). However, NO concentrations in the bottom layer showed negative correlations with salinity ($p < 0.05$, $R = -0.258$) and

bottom depth ($p < 0.05$, $R = -0.292$) indicating a decrease of NO concentrations from the coast toward offshore waters. It is known that both the BS and YS are affected by submarine ground water discharge (Kim et al., 2005; Liu et al., 2017a; 2017b; Taniguchi et al. 2008). Moreover, NO has been detected in groundwater and aquifers (Smith et al., 2004, Smith and Yoshinari, 2008). Therefore, we suggest that input of NO from submarine groundwater discharge contributed to the distribution of NO bottom concentrations as well.

An overview of published NO surface concentrations is given Table 3. The average NO surface and bottom concentrations from this study are comparable to the concentrations measured in the Jiaozhou Bay, in the waters off Qingdao and in the Seto Inland Sea. However, our mean concentrations are considerably higher than the NO concentrations reported from the central equatorial Pacific and then eastern tropical North Pacific Ocean. The maximum concentration reported here is at the lower end of the NO concentrations recently reported from the anoxic oxygen minimum zone off Peru. Overall, NO surface

concentrations NO seem to be generally higher in coastal waters compared to those found in offshore waters.

### 3.2 Diurnal variability

The diurnal variability of surface NO concentrations, illuminance, DO, and Chl-*a* were investigated at the anchor station B65 (Fig. 4). NO concentrations varied from 64 to 424 pmol $L^{-1}$, exhibiting a significant diurnal variation with the maximum concentration eightfold higher than the minimum concentration. The NO concentrations reached the maximum concentration

in the early afternoon (about 13:00 local time, LT) and then decreased to the minimum concentration at 22:00 LT. A less pronounced second maximum (209 pmol $L^{-1}$) was reached at 04:00 LT. DO showed a similar diurnal cycle but shifted by three hours with maxima at 16:00 LT and 07:00 LT. Chl-*a* concentrations peaked at 19:00 LT and 07:00 LT. The illuminance had its maximum at 13:00 LT coinciding with the NO maximum indicating that the first NO maximum was indeed resulting from a photochemical production during daytime (Zafiriou and McFarland, 1981). However, the second

maximum of NO at 04:00h, when it was still dark, must have resulted from an alternative chemical and/or biological production.

### 3.3 Photoproduction rates

The results of the NO photoproduction experiments are listed in Table 2. The photoproduction rates of NO in the BS, the NYS, and the SYS varied from 0.00 to $5.07 \times 10^{-11}$ mol $L^{-1} s^{-1}$, 0.09 to $0.69 \times 10^{-11}$ mol $L^{-1} s^{-1}$, and 0.32 to $1.54 \times 10^{-11}$ mol $L^{-1} s^{-1}$,

respectively. The average photoproduction rate of the whole study area was $1.14 \pm 1.37 \times 10^{-11}$ mol $L^{-1} s^{-1}$. The photoproduction rates from the BS and YS are in good agreement with the rates reported from Seto Inland Sea (Olasehinde et al, 2009; 2010). However, the mean NO photoproduction rates are higher than those from the central equatorial Pacific



Ocean (Zafiriou and McFarland, 1981) and lower than those from Kurose River, Japan (Olasehinde et al, 2009) (Table 2) which mirrors the available $NO_2^-$ concentrations which are low in the open ocean and but high in a river.

The average photoproduction rate in the NYS was obviously lower than those in the BS and the SYS, consistent with the average illuminances, which were 22450, 20433, and 27825 lx for the BS, the NSY and the SYS, respectively. Enhanced photoproduction rates ($\geq 2.00 \times 10^{-11}$ mol $L^{-1}$ $s^{-1}$) occurred in the SYS, especially in the central part of the southern SYS and stations influenced by the Yellow Sea Cold Current (see Fig. 1). This is apparently in contrast to the distribution of NO which showed lowest concentrations in the central SYS (see above). The high illumination observed in the SYS (Table 2) does not only lead to enhanced NO photoproduction (see Table 2) but also generate reactive oxygen species like $O_2\bullet-$, $ROO\bullet$, and other OH-derived radicals, which in turn can efficiently scavenge NO (Olasehinde et al, 2010). Overall, the NO photoproduction rates showed a positive relationship with illuminance ($p < 0.01$, $R = 0.884$) indicating that the NO concentrations in the surface layer during daylight was dominated by photochemical production. However, we did not find a significant difference between the mean NO concentrations sampled during day ($179 \pm 80$ pmol $L^{-1}$) and night ($195 \pm 140$ pmol $L^{-1}$). This suggests that there was also a non-photochemical NO dark production in the surface layer.

### 3.4 Sea-to-air fluxes of NO

In the present study, the NO supersaturation was ubiquitous at all investigated sites. The supersaturation factors varied from 8 to 154, with an average of 47. Together with the fact that that NO was supersaturated during daytime and nighttime during the 24h station B65 this indicates that the BS and YS were a source of NO to the atmosphere. The sea-to-air flux densities ranged from $5.8 \times 10^{-19}$ to $3.6 \times 10^{-15}$ mol $cm^{-2}$ $s^{-1}$, with an average value of $4.5 \times 10^{-16}$ mol $cm^{-2}$ $s^{-1}$ (Fig. 7). The comparison of wind speeds and flux densities reveal that the flux densities are mainly driven by the wind speed (Fig. 7) Our flux densities from the BS and YS were similar to those computed for the central equatorial Pacific Ocean and the Seto Inland Sea, while they are slightly lower than those computed for of Jiaozhou Bay waters (Table 2). Based on the YS area of $380 \times 10^3$ $km^2$ and the BS area of $77 \times 10^3$ $km^2$, the emission of NO to the atmosphere was estimated to be $9.0 \times 10^8$ g N $yr^{-1}$ or $7.6 \times 10^7$ g N $month^{-1}$. Ding et al. (2018) report an satellite-derived average $NO_x$ emission estimate for June over the BS and the YS in the period from 2007 to 2016 of about $1.3 \times 10^{10}$ g N $month^{-1}$. The obvious very large discrepancy between the satellite-derived emission estimate and the one presented here results from the fact that Ding et al.'s (2018) estimate is dominated by the $NO_x$ emissions from ships' diesel engines. This indicates that oceanic NO emissions to the atmosphere only account for a negligible fraction (~0.6 %) of the $NO_x$ emissions observed over the BS and YS.

### 3.5 NO mixed layer budget

In order to estimate the contribution of different source and sinks of NO in surface layer of the BS and YS we applied a simple box model. We assume that the surface layer is represented by the mixed layer with a mean water depth of 15 m depth (Qiao et al., 2004). At steady state the loss of NO by air-sea exchange ($F_{ase}$) must be equal to the sum of the photoproduction rate ($F_{pp}$) in the mixed layer and the input from below into the mixed layer by diapycnal diffusion ($F_{dia}$) and





other production or consumption pathways ($F_{poc}$) and advection into or out of the BS/YS mixed layer ($F_{adv}$). To this end, $F_{ase}$ is given by

$$F_{ase} = F_{pp} + F_{dia} + F_{poc} + F_{adv}.$$

The mean $F_{ase}$ was $4.5 \times 10^{-16}$ mol cm$^{-2}$ s$^{-1}$ (see above). We assume that the NO photoproduction decreases linearly from the average rate of $1.15 \pm 1.47 \times 10^{-11}$ mol L$^{-1}$ s$^{-1}$ at the surface (see above) to 0 at 15 m. The mean $F_{pp}$ was calculated to be $8.6 \times 10^{-12}$ mol cm$^{-2}$ s$^{-1}$ in mixed layer. Since the mean surface NO concentration in the surface layer was statistically not different from the mean NO concentration in the bottom layer (see above) it is reasonable to assume that $F_{dia} = 0$. Moreover, it seems reasonable to assume that advection of NO into or out of the BS/YS surface layer is zero. Since $F_{ase} < F_{pp}$ we conclude that $F_{poc}$ should be negative indicating that the bulk of the produced NO was rapidly consumed in the surface layer before its release to the atmosphere. Please note that we have some indications that there is also a dark production of NO (see above), so that the 'true' NO surface production might be even higher. Chemical reactions with DO, OH, or ROO• etc. are potential sinks for NO in the surface layer of the BS and YS (Ford et al., 1993; Olasehinde et al, 2010; Carpenter and Nightingale, 2015).

## 4. Conclusions

This study reports the distribution and photoproduction rates of dissolved NO measured during a cruise in June 2011 to the Bohai and Yellow Seas. The NO concentrations in both the surface and bottom layers were highly variable. There was no significant difference between the mean NO concentrations in the surface and bottom layers. NO concentrations in the bottom layer showed significant decreasing with salinity indicates NO input by submarine groundwater disharge. NO in the surface layer was supersaturated at all stations during day and night. The accumulation of NO during daytime was resulting from photoproduction and the measured NO photoproduction rates were correlated to illuminance. The persistent nighttime NO supersaturation pointed to a, so far unknown, non-photochemical (chemical and/or biological) NO dark production. On the basis of a simple box model calculation we conclude that the bulk of the NO produced in the surface layer was rapidly consumed before its release to the atmosphere. Overall, the BS and YS were a persistent source of NO to the atmosphere at the time of our measurements. However, the oceanic NO emissions were negligible compared to the NO$_x$ emissions from ships diesel engines.

**Author contribution**: Chun-Ying Liu and Gui-Peng Yang designed the experiments and Ye Tian, Chao Xue, Pei-Feng Li, and Wei-Hua Feng carried them out. Hermann W. Bange and Chun-Ying Liu analyzed the data and Ye Tian prepared the manuscript with contributions from all co-authors.

**Competing interests.** The authors declare that they have no conflict of interest.





## Acknowledgements

We thank the captain and crew of the R/V "Dong Fang Hong 2" for their support and help during the cruise. This research was supported by the National Natural Science Foundation of China (Nos.41676065 and 40706040), the National Key Research and Development Program of China (Grant No. 2016YFA0601301), the Fundamental Research Funds for the

Central Universities (No. 201762032), and the AoShan Talents Program of Qingdao

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

**Figure Captions**

**Figure 1.** Locations of the sampling stations in the BS and the YS during summer.

**Figure 2.** Horizontal distributions of NO (pmol Ł$^{-1}$) in the surface water and bottom water.

5 **Figure 3.** Variations of salinity, NO surface concentrations, and $NO_2^-$ concentrations from station B65 to station B70.

**Figure 4.** Diurnal variations of NO concentrations, illuminance (I), DO, and Chl-*a* concentrations in the surface water at the

anchor station B65.

**Figure 5.** Wind speeds and flux densities of NO in the Bohai and Yellow Seas.

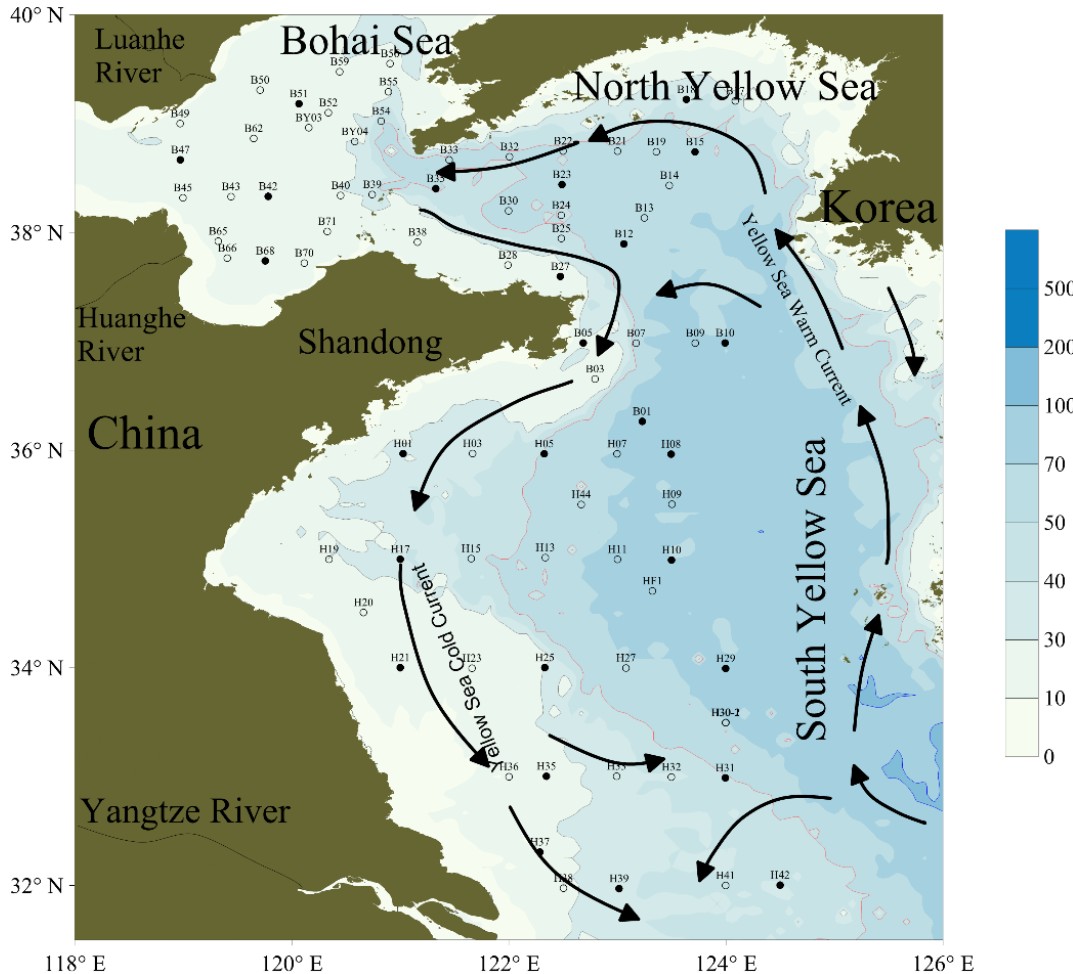

10 **Figure 1.** Locations of the sampling stations in the BS and the YS during summer.
Solid dots (●) represent the stations for incubation experiments.

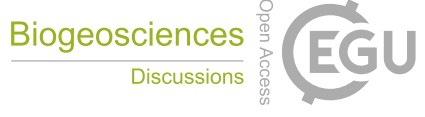

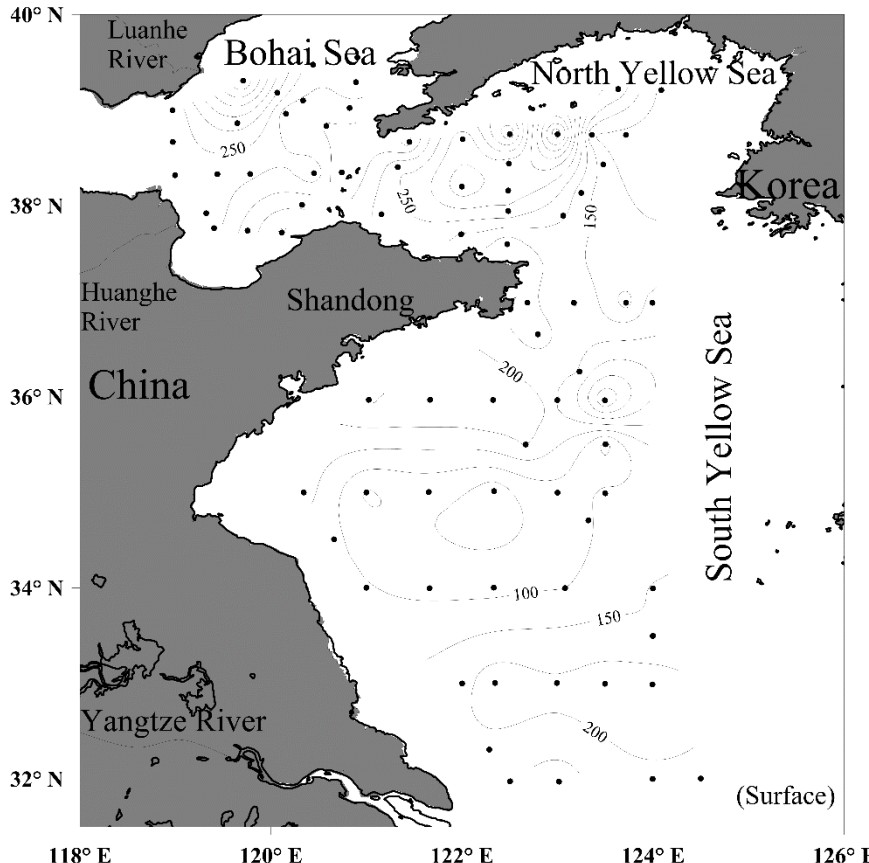




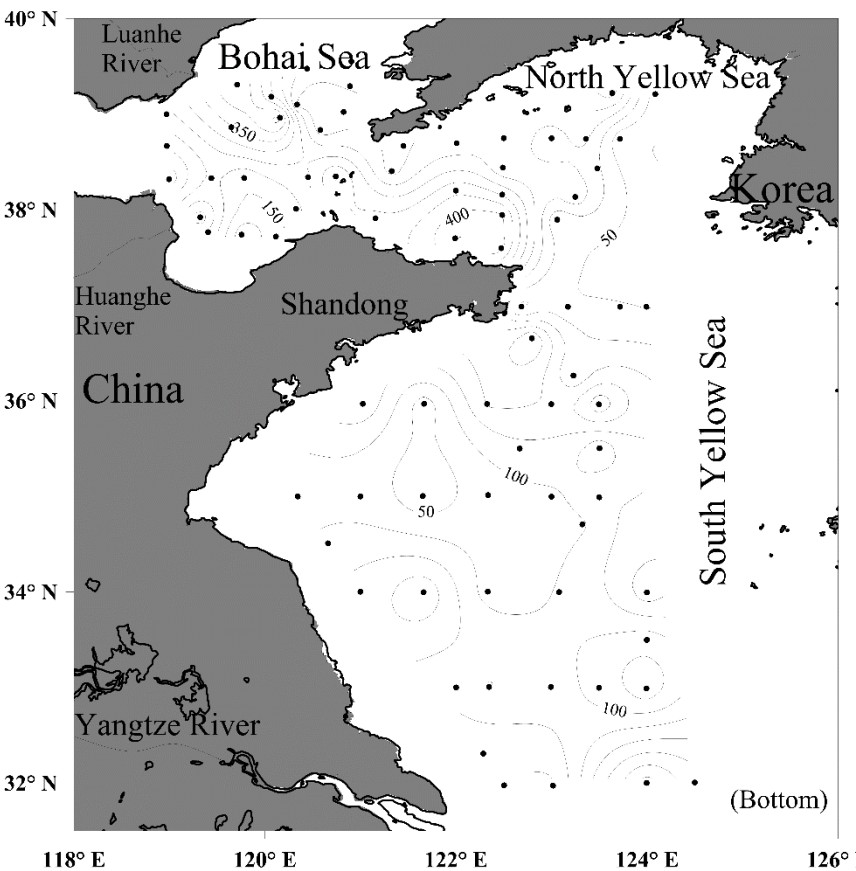

**Figure 2.** Horizontal distributions of NO (pmol Ł$^{-1}$) in the surface and bottom layers.




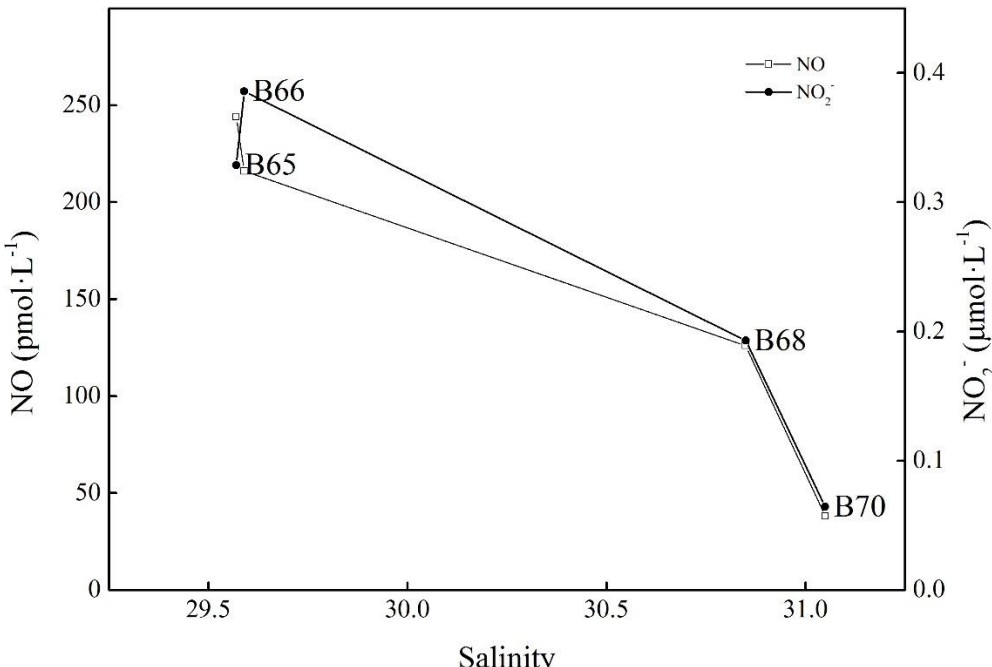

**Figure 3.** Variations of salinity, NO surface concentrations, and $NO_2^-$ concentrations from station B65 to station B70.





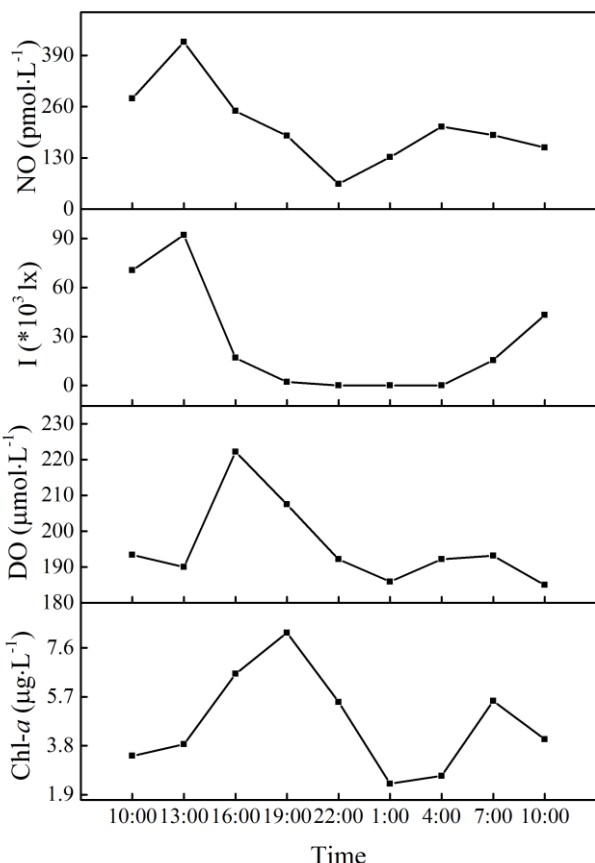

**Figure 4.** Diurnal variations of NO concentrations, illuminance (I), DO, and Chl-*a* concentrations in the surface water at the anchor station B65.



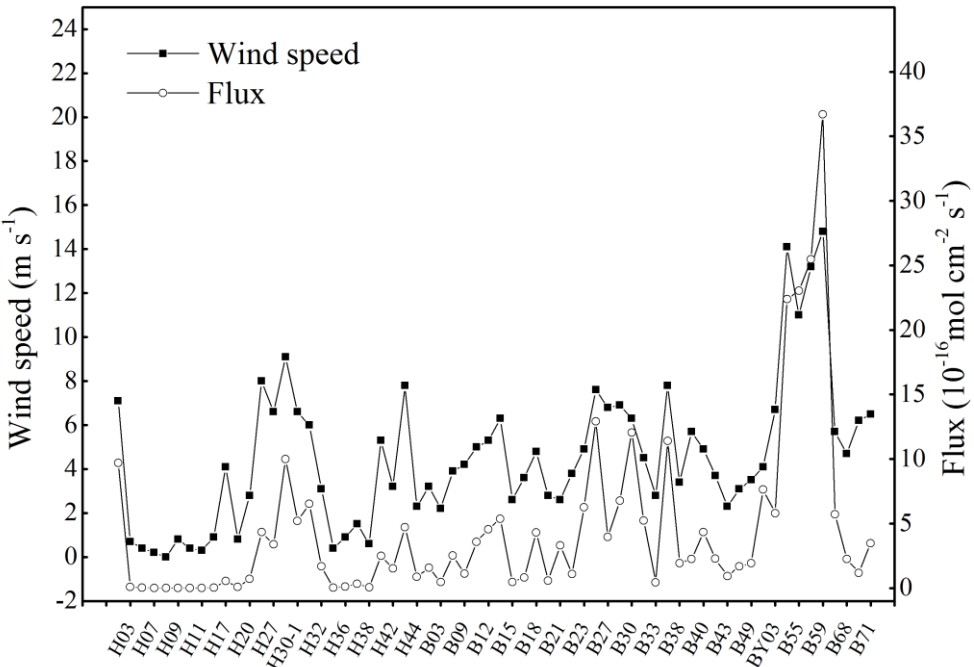

**Figure 5.** Wind speeds and flux densities of NO from the Bohai and Yellow Seas.

**Table Captions**

**Table 1** Description of sampling stations and seawater temperature (T), salinity (S), Chl-*a* concentrations, DO, illuminance

(I), and NO concentrations in the BS and the YS.

**Table 2** Description of sampling stations and their seawater temperature, illumination intensity, and photoproduction rates in

5    the BS and the YS.

**Table 3** NO Concentrations of different regions in literatures.



**Table 1** Description of sampling stations and seawater temperature (T), salinity(S), Chl-*a* concentrations,

| Station | Location | Depth (m) | Local time | $T_{surf}$ (℃) | $T_{bott}$ (℃) | $S_{Surf}$ ‰ | $S_{bott}$ ‰ | Chl-$a_{surf}$ (µg L⁻¹) | $DO_{surf}$ (µM) | $DO_{bott}$ (µM) | I (lx) | $[NO]_{surf}$ (pM) | $[NO]_{bott}$ (pM) |
|---|---|---|---|---|---|---|---|---|---|---|---|---|---|
| H01 | 121.03 °E, 35.97 °N | 33 | 1928 | 15.0 | 9.7 | 31.72 | 31.76 | 0.565 | 233.35 | 265.84 | 169 | 257 | 196 |
| H03 | 121.67 °E, 35.97 °N | 37 | 2250 | 18.3 | 8.8 | 31.73 | 31.68 | 0.726 | 212.8 | 242.69 | 32 | 248 | <LOD |
| H05 | 122.33 °E, 35.97 °N | 55 | 0238 | 17.1 | 6.3 | 31.50 | 32.25 | 0.207 | 231.82 | 203.8 | 32 | 253 | 206 |
| H07 | 123.00 °E, 35.97 °N | 71 | 0618 | 17.0 | 7.7 | 31.55 | 32.93 | 0.446 | 226.54 | 199.21 | 19090 | 195 | 195 |
| H08 | 123.50 °E, 35.96 °N | 75 | 0929 | 17.1 | 7.7 | 32.35 | 32.99 | 7.674 | 222.67 | 194.6 | 57700 | 407 | 295 |
| H09 | 123.50 °E, 35.50 °N | 76 | 1240 | 18.3 | 7.8 | 32.20 | 33.05 | 7.284 | 215.41 | 202.61 | 101500 | 32 | 57 |
| H10 | 123.50 °E, 34.99 °N | 77 | 1546 | 18.7 | 7.8 | 32.53 | 33.16 | 3.206 | 218.32 | 197.55 | 13200 | 110 | 148 |
| H11 | 123.00 °E, 35.00 °N | 72 | 1842 | 19.1 | 8.1 | 31.97 | 33.36 | 2.587 | 220.06 | 196.44 | 58 | 90 | 135 |
| H13 | 122.34 °E, 35.01 °N | 62 | 2242 | 19.0 | 7.7 | 31.67 | 32.87 | 0.521 | 215.07 | 199.33 | 32 | <LOD | 78 |
| H15 | 121.65 °E, 35.00 °N | 46 | 0235 | 19.6 | 6.3 | 31.59 | 32.00 | 0.627 | 207.00 | 213.24 | 32 | 35 | <LOD |
| H17 | 121.00 °E, 35.00 °N | 38 | 0612 | 18.4 | 10.5 | 31.19 | 31.86 | 2.400 | 224.96 | 229.62 | 33900 | 41 | 81 |
| H19 | 120.34 °E, 35.00 °N | 28 | 1013 | 17.2 | 15.3 | 31.62 | 31.71 | 1.375 | 220.03 | 218.6 | 98700 | 172 | 55 |
| H20 | 120.66 °E, 34.51 °N | 20 | 1354 | 19.0 | 18.8 | 30.01 | 30.00 | 0.470 | 216.86 | 207.34 | 77400 | 113 | 122 |
| H21 | 121.00 °E, 34.00 °N | 19 | 1733 | 20.4 | 20.1 | 31.00 | 30.88 | 0.597 | 191.46 | 194.87 | 6490 | NA | NA |
| H23 | 121.66 °E, 34.00 °N | 20 | 2104 | 19.1 | 19.1 | 31.96 | 31.96 | 4.417 | 197.10 | 196.78 | 32 | NA | 244 |
| H25 | 122.33 °E, 34.00 °N | 40 | 0112 | 19.3 | 9.8 | 32.00 | 32.51 | 5.632 | 211.41 | 225.14 | 32 | 86 | 78 |
| H27 | 123.08 °E, 34.00 °N | 70 | 0618 | 18.9 | 8.8 | 32.73 | 33.36 | 4.938 | 205.1 | 204.93 | 10122 | 99 | 104 |
| H29 | 124.00 °E, 33.99 °N | 82 | 1257 | 19.2 | 9.1 | 32.63 | 33.71 | 13.933 | 208.46 | 180.25 | 65183 | 153 | 232 |
| H30-1 | 124.00 °E, 33.50 °N | 69 | 1615 | 18.4 | 10.1 | 31.79 | 33.65 | 1.047 | 226.07 | 198.98 | 12780 | 153 | 70 |
| H31 | 123.99 °E, 32.99 °N | 49 | 1912 | 18.6 | 11.9 | 31.68 | 32.72 | 1.593 | 222.42 | 191.59 | 11330 | 231 | <LOD |
| H32 | 123.50 °E, 33.00 °N | 39 | 2130 | 18.4 | 13.8 | 31.65 | 32.27 | 15.783 | 229.45 | 191.93 | 32 | 226 | NA |
| H33 | 122.99 °E, 33.00 °N | 36 | 2352 | 17.6 | 14.8 | 31.91 | 32.15 | 7.636 | 224.52 | 198.59 | 32 | NA | 124 |
| H35 | 122.35 °E, 33.00 °N | 36 | 0358 | 17.6 | 17.6 | 31.69 | 31.71 | 3.195 | 210.63 | 211.31 | 32 | 233 | 154 |
| H36 | 122.00 °E, 33.00 °N | 14 | 0610 | 18.4 | 18.4 | 31.98 | 31.99 | 2.272 | 198.29 | 197.87 | 44800 | 183 | 102 |
| H37 | 122.29 °E, 32.31 °N | 25 | 1037 | 19.1 | 19.0 | 30.95 | 31.08 | 5.725 | 181.50 | 180.75 | 22200 | 189 | 68 |
| H38 | 122.50 °E, 31.97 °N | 27 | 1328 | 19.2 | 18.3 | 29.65 | 31.28 | 6.407 | 177.64 | 147.26 | 9620 | 179 | 237 |
| H39 | 123.02 °E, 31.97 °N | 38 | 1600 | 18.6 | 18.5 | 31.89 | 31.89 | 3.406 | 180.44 | 178.31 | 18140 | 114 | 36 |
| H41 | 124.00 °E, 32.00 °N | 43 | 2125 | 17.2 | 17.1 | 31.41 | 31.45 | 1.090 | 202.38 | 201.91 | 32 | NA | 376 |
| H42 | 124.50 °E, 32.00 °N | 43 | 0030 | 18.7 | 15.4 | 31.90 | 32.24 | 0.897 | 225.02 | 199.71 | 32 | 188 | 271 |
| HF1 | 123.32 °E, 34.71 °N | 78 | 1635 | 20.3 | 8.2 | 32.38 | 33.40 | 0.671 | 203.16 | 191.03 | 10940 | 97 | 92 |





| H44 | 122.67 °E, 35.50 °N | 69 | 2348 | 19.7 | 7.6 | 31.51 | 32.91 | 3.712 | 222.87 | 185.19 | 32 | 205 | NA |
| B01 | 123.23 °E, 36.26 °N | 75 | 0328 | 16.9 | 7.3 | 31.56 | 32.80 | 0.605 | 240.44 | 204.71 | 528 | 201 | 95 |
| B03 | 122.79 °E, 36.65 °N | 25.7 | 0753 | 15.4 | 11.7 | 31.38 | 31.48 | 0.236 | 258.34 | 243.82 | 54200 | 126 | 311 |
| B05 | 122.69 °E, 36.98 °N | 41 | 1136 | 15.5 | 11.5 | 31.59 | 31.62 | 1.494 | 247.11 | 235.10 | 100000 | NA | <LOD |
| B07 | 123.17 °E, 36.98 °N | 63 | 1443 | 20.5 | 6.7 | 31.57 | 32.23 | 0.309 | 222.31 | 237.81 | 76500 | 208 | 60 |
| B09 | 123.72 °E, 36.98 °N | 76 | 1811 | 20.0 | 6.8 | 31.92 | 32.42 | 0.441 | 211.24 | 236.46 | 2440 | 79 | NA |
| B10 | 123.99 °E, 36.98 °N | 77 | 1947 | 21.0 | 6.8 | 31.95 | 32.42 | 0.312 | 204.13 | 227.81 | 922 | 177 | 48 |
| B12 | 123.06 °E, 37.89 °N | 62 | 0200 | 18.8 | 6.4 | 31.64 | 32.14 | 0.149 | 217.03 | 235.66 | 32 | 207 | 49 |
| B13 | 123.25 °E, 38.13 °N | 65 | 0440 | 18.9 | 5.9 | 31.73 | 32.09 | 0.398 | 212.71 | 257.77 | 203 | 172 | 191 |
| B14 | 123.48 °E, 38.43 °N | 66 | 0700 | 19.2 | 7.1 | 31.76 | 32.15 | 0.427 | 218.36 | 262.51 | 2500 | NA | NA |
| B15 | 123.72 °E, 38.74 °N | 59 | 0918 | 19.4 | 8.7 | 31.79 | 32.11 | 0.274 | 216.53 | 262.56 | 16640 | 90 | <LOD |
| B17 | 124.09 °E, 39.21 °N | 41 | 1300 | 11.3 | 10.9 | 31.62 | 31.65 | 0.663 | 260.47 | 259.75 | 12930 | 89 | <LOD |
| B18 | 123.64 °E, 39.22 °N | 49 | 1528 | 19.7 | 8.7 | 31.39 | 31.84 | 0.600 | 223.91 | 256.77 | 5140 | 236 | 276 |
| B19 | 123.36 °E, 38.74 °N | 57 | 1852 | 19.7 | 7.3 | 31.57 | 32.13 | 2.276 | 216.04 | 264.95 | 1010 | 94 | 161 |
| B21 | 123.00 °E, 38.75 °N | 54 | 2052 | 19.7 | 5.2 | 31.29 | 31.98 | 0.527 | 214.72 | 264.49 | 32 | 616 | 123 |
| B22 | 122.50 °E, 38.75 °N | 55 | 2348 | 17.4 | 3.9 | 30.81 | 32.03 | 1.067 | 246.19 | 249.64 | 32 | 98 | 192 |
| B23 | 122.49 °E, 38.44 °N | 55 | 0210 | 20.1 | 4.9 | 31.66 | 31.93 | 0.579 | 207.68 | 245.58 | 32 | 327 | 154 |
| B25 | 122.48 °E, 37.94 °N | 49 | 1616 | 18.4 | 5.2 | 31.17 | 31.79 | 0.090 | 224.02 | 239.04 | 8720 | 286 | 407 |
| B27 | 122.47 °E, 37.60 °N | 27 | 1903 | 18.0 | 14.1 | 31.16 | 31.47 | 0.369 | 229.34 | 188.49 | 203 | 110 | 417 |
| B28 | 121.99 °E, 37.70 °N | 22.8 | 2143 | 13.3 | 9.4 | 31.47 | 31.60 | 2.080 | 270.02 | 247.82 | 32 | 194 | 453 |
| B30 | 122.00 °E, 38.20 °N | 56 | 0147 | 18.7 | 3.8 | 31.18 | 32.02 | 1.128 | 217.62 | 246.15 | 32 | 387 | 338 |
| B32 | 122.01 °E, 38.70 °N | 53 | 0550 | 16.1 | 4.6 | 30.61 | 31.91 | 4.697 | 240.34 | 251.92 | 25000 | 339 | 171 |
| B33 | 121.43 °E, 38.67 °N | 61 | 0752 | 18.1 | 5.6 | 30.82 | 31.81 | 2.737 | 221.37 | 252.02 | 28700 | 72 | NA |
| B35 | 121.33 °E, 38.40 °N | 50 | 1052 | 17.0 | 6.3 | 30.81 | 31.75 | 4.616 | 237.4 | 252.03 | 30900 | 244 | 128 |
| B38 | 120.74 °E, 38.35 °N | 22 | 1530 | 15.2 | 12.1 | 31.19 | 31.47 | 1.945 | 251.83 | 255.95 | 15000 | 223 | 315 |
| B39 | 120.45 °E, 38.34 °N | 29 | 2021 | 13.8 | 12.3 | 31.19 | 31.30 | 0.706 | 241.53 | 240.32 | 32 | 94 | 363 |
| B40 | 119.78 °E, 38.33 °N | 30 | 2208 | 14.4 | 12.1 | 30.97 | 31.28 | 0.382 | 254.71 | 245.62 | 32 | 243 | 241 |
| B42 | 119.44 °E, 38.33 °N | 26.8 | 0550 | 19.0 | 12.6 | 30.64 | 31.37 | 1.151 | 214.26 | 216.44 | 19700 | 211 | 157 |
| B43 | 119.00 °E, 38.32 °N | 24 | 0740 | 18.8 | 13.9 | 30.93 | 31.32 | 2.667 | 226.67 | 188.85 | 42400 | 224 | 208 |
| B45 | 118.97 °E, 38.67 °N | 20.5 | 1022 | 19.4 | 17.2 | 31.07 | 31.22 | 1.612 | 236.45 | 208.28 | 81900 | NA | 60 |
| B47 | 118.97 °E, 38.67N | 25 | 1325 | 17.4 | 13.0 | 31.18 | 31.47 | 0.133 | 251.91 | 212.27 | 91900 | 227 | NA |
| B49 | 118.97 °E, 39.00 °N | 21.4 | 1610 | 17.6 | 16.3 | 31.31 | 31.37 | 0.596 | 270.9 | 240.59 | 29000 | 204 | 236 |
| B50 | 119.71 °E, 39.31 °N | 26 | 2010 | 20.7 | 11.8 | 31.25 | 31.37 | 0.135 | 215.16 | 212.47 | 32 | 565 | 452 |





| | | | | | | | | | | | | | |
|---|---|---|---|---|---|---|---|---|---|---|---|---|---|
| B51 | 120.07 °E, 39.18 °N | 24 | 2220 | 19.3 | 14.8 | 31.21 | 31.31 | 0.625 | 220.94 | 213.47 | 32 | NA | 307 |
| BY03 | 120.16 °E, 38.96 °N | 21 | 0008 | 17.7 | 17.5 | 31.15 | 31.15 | 0.843 | 242.26 | 246.04 | 32 | 167 | 482 |
| B52 | 120.34 °E, 39.10 °N | 22 | 0201 | 18.6 | 17.6 | 31.23 | 31.22 | 1.114 | 245.12 | 231.72 | 32 | NA | 141 |
| BY04 | 120.58 °E, 38.84 °N | 37 | 0515 | 16.8 | 15.8 | 31.24 | 31.27 | 0.423 | 218.19 | 206.65 | 1463 | 147 | 125 |
| B55 | 120.89 °E, 39.29 °N | 33 | 0929 | 16.9 | 13.2 | 31.16 | 31.36 | 0.210 | 231.28 | 226.55 | 39500 | 248 | NA |
| B56 | 120.91 °E, 39.55 °N | 31.6 | 1124 | 17.5 | 14.0 | 31.29 | 31.40 | 0.967 | 214.96 | 191.25 | 70500 | 189 | 204 |
| B59 | 120.44 °E, 39.48 °N | 27 | 1357 | 17.4 | 12.7 | 31.25 | 31.41 | 0.232 | 215.45 | 192.18 | 86200 | 217 | 185 |
| B65 | 119.65 °E, 38.86 °N | 16 | 0957 | 20.8 | NA | 29.57 | NA | 0.323 | 193.56 | NA | 31800 | 244 | 335 |
| B66 | 119.32 °E, 37.92 °N | 14 | 1225 | 21.7 | 21.6 | 29.59 | 30.03 | 0.493 | 190.04 | 174.75 | 35500 | 216 | 244 |
| B68 | 119.41 °E, 37.73N | 16.6 | 1442 | 21.5 | 21.1 | 30.85 | 30.86 | 0.757 | 195.03 | 179.67 | 30000 | 126 | 63 |
| B70 | 120.12 °E, 37.72N | 17 | 1717 | 20.7 | 19.1 | 31.05 | 31.17 | 0.988 | 202.73 | 187.42 | 109 | 38 | 131 |
| B71 | 120.22 °E, 38.01N | 19.9 | 1939 | 17.8 | 17.2 | 31.18 | 31.19 | 0.344 | 226.72 | 212.02 | 53 | 106 | 184 |

NA: not available
<LOD: below the limit of detection





**Table 2** Description of sampling stations and their seawater temperature, illuminance, and photoproduction rates in the BS and the YS.

|  | Station | I (lx) | T (°C) | photoproduction rate ($10^{-11}$ mol L$^{-1}$s$^{-1}$) |
|---|---|---|---|---|
| SYS | H01 | 7310 | 15.0 | 0.21 |
|  | H05 | 57700 | 17.1 | 3.18 |
|  | H08 | 57700 | 17.1 | 2.92 |
|  | H10 | 57700 | 18.7 | 5.07 |
|  | H17 | 57700 | 18.4 | 2.79 |
|  | H21 | 57700 | 20.4 | 4.37 |
|  | H25 | 16790 | 19.3 | 1.28 |
|  | H29 | 16790 | 19.2 | 0.00 |
|  | H31 | 16790 | 18.6 | 0.51 |
|  | H35 | 16790 | 17.6 | 0.22 |
|  | H37 | 16790 | 19.1 | 1.17 |
|  | H39 | 18140 | 18.6 | 1.29 |
|  | H42 | 18140 | 18.7 | 0.16 |
|  | B01 | 9720 | 16.9 | 0.28 |
|  | B05 | 9720 | 15.5 | 0.05 |
|  | B10 | 9720 | 21.0 | 0.09 |
|  | Average | 27825 | 18.2 | 1.47 |
| NYS | B12 | 15200 | 18.8 | 0.45 |
|  | B15 | 15200 | 19.4 | 0.09 |
|  | B18 | 15200 | 19.7 | 0.22 |
|  | B23 | 15200 | 20.1 | 0.69 |
|  | B27 | 30900 | 18.0 | 0.27 |
|  | B35 | 30900 | 17.0 | 0.10 |
|  | Average | 20433 | 18.3 | 0.33 |
| BS | B42 | 30900 | 19.0 | 1.54 |
|  | B47 | 14000 | 17.4 | 0.88 |
|  | B51 | 14000 | 19.3 | 0.32 |
|  | B68 | 30900 | 21.5 | 1.50 |
|  | Average | 22450 | 19.3 | 1.14 |



**Table 3** NO concentrations and flux densities from different regions.

| Regions | [NO] (pmol L$^{-1}$) | NO fluxes (mol cm$^{-2}$ s$^{-1}$) | Sampling date | Sampling depth | Reference |
|---|---|---|---|---|---|
| Jiaozhou Bay | 157 | $7.2 \times 10^{-16}$ | June, July and August, 2010 | Surface water | Tian et al., 2015 |
| Jiaozhou Bay and its adjacent waters | 160 ±130 | $10.9 \times 10^{-16}$ | March 8-9, 2011 | Surface water | Xue et al., 2012 |
| Central equatorial Pacific | 46 | $> 2.2 \times 10^{-16}$ | July 14 to August 16, 1978 | Surface water | Zafiriou and McFarland, 1981 |
| Eastern tropical North Pacific Ocean | 0-65 | - | November, 1983 | 0-3500m | Ward and Zafiriou, 1988 |
| Eastern tropical South Pacific Ocean off Peru | <500-9500 | - | February 6 to March 11, 2013 | Surface-327 m | Lutterbeck et al., 2018 |
| Coastal water off Qingdao | 260 ±140 | - | November, 2009 | Surface water | Liu et al.,2017 |
| Seto Inland Sea, Japan | 24-320 | $3.55 \times 10^{-16}$ | October 5-9, 2009 | Surface water | Olasehinde et al, 2010 |
| Yellow Sea and Bohai Sea | Surface: 186 ±108 Bottom: 174 ±123 | $4.5 \times 10^{-16}$ | June 13-28, 2011 | 1 m and 1 m above the bottom | This study |