# Peer review of "Nitric oxide (NO) in the Bohai and Yellow Seas"

_Biogeosciences, 2018_

## Referee Comment (RC1) · Anonymous Referee #1 · 8 Nov 2018

Manuscript title: Nitric oxide (NO) in the Bohai and Yellow Seas

The reviewer found the manuscript acceptable since it provided relevant findings. However, the following observations are pointed out: 1. The last sentence in the abstract seems spurious and unsubstantiated, because there was no part of the manuscript that reported research finds on NOx emissions from ships 2. Judging from the distribution patterns of NO flux and the wind speed (Figure 5), one would expect an important inference and conclusive statement in the abstract and discussion. Previous investigators (Anifowose AJ, Sakugawa H. 2017. Determinaton of Daytme Flux of Nitric Oxide Radical (NOâ Ăć) at an Inland Sea-Atmospheric Boundary in Japan. J Aquat Pollut Toxicol., 1:2) reported wind speed as an important factor governing NO flux at the air-sea interface. 3. The analytical methods on the measurements of NO concentration and during irradiation experiments (photoformation rate) are not explicit enough. I understand that the authors referred to Liu et al. (2017), there is need for them to report detailed an-

alytical procedures in the manuscript. 4. In view of comment 3 above, one would ask the precious question as to whether the measured concentration of NO during the irradiation experiment was steady-state concentration, even when the NO scavenging rate constant in the seawater (during the experiment) remained unknown? 5. I think there should be comprehensive correlation plot (and its discussion in the manuscript) of relationship between NO and NO2  (a major NO source). While it is true that the authors presented Figure 3 to reflect this, we only have very scanty data plotted. 6. Page 2, Line 1: 10 % should be 10%. This should be applicable in other relevant places in the manuscript. 7. Page 3: we have interchangeable use of "h" and "hour(s)". The authors should stick to "h" preferably. 8. Page 6, Line 29: "0.00 × 10-11 molL-1s-1" should be "not detectable" 9. Page 7: The statement between Lines 8 and 9 should read "…but would also generate reactive oxygen species like O2 -, ROO  and other OH-related radicals, which in turn, would efficiently scavenge NO…" 10. Page 7, Line 17: 24h should be 24 h 11. Page 10, Line 19: 13(4), 1-31 were repeated in the reference

---

## Author Comment (AC1) · 18 Nov 2018

1. The last sentence in the abstract seems spurious and unsubstantiated, because there was no part of the manuscript that reported research finds on NOx emissions from ships.

In our manuscript of Page 7 line 20-25, we compared the NO emission of the Yellow Sea and the Bohai Sea to the atmosphere based on our observation ($7.6 \times 10^7$ gN•month-1) with the satellite-derived average NOx emission estimate (about $1.3 \times 10^{10}$ g N•month-1) reported by Ding et al., 2018. Because Ding et al.'s (2018) estimate is dominated by the NOx emissions from ships' diesel engines, thus we concluded that the oceanic NO emissions to the atmosphere were negligible compared to anthropogenic NOx sources such as emissions from ships.

2. Judging from the distribution patterns of NO flux and the wind speed (Figure 5),

one would expect an important inference and conclusive statement in the abstract and discussion. Previous investigators (Anifowose AJ, Sakugawa H. 2017. Determinaton of Daytme Flux of Nitric Oxide Radical (NOâAËŸ c) at an Inland Sea-Atmospheric Boundary in Japan. J Aquat Pollut Toxicol., ′ 1:2) reported wind speed as an important factor governing NO flux at the air-sea interface.

Yes, wind speed was an important factor governing NO flux at the air-sea interface judging from the distribution patterns of NO flux and the wind speed. We will supplement it in the abstract and discussion of revised manuscript and cite this reference of Anifowose and Sakugawa (2017).

3. The analytical methods on the measurements of NO concentration and during irradiation experiments (photoformation rate) are not explicit enough. I understand that the authors referred to Liu et al. (2017), there is need for them to report detailed analytical procedures in the manuscript.

Thank you for your comment. We will supplement the analytical methods on the measurements of NO concentration and during irradiation experiments in revised manuscript.

4. In view of comment 3 above, one would ask the precious question as to whether the measured concentration of NO during the irradiation experiment was steady-state concentration, even when the NO scavenging rate constant in the seawater (during the experiment) remained unknown?

During the irradiation of sunlight, production and consumption of NO occurs simultaneously in seawater bulk, for example, radicals from CDOM can scavenge NO, and thus the measured concentration of NO was a net value of NO production.

5. I think there should be comprehensive correlation plot (and its discussion in the manuscript) of relationship between NO and $NO_2$âA£ (a major NO source). While it is true that the authors presented Figure 3 to reflect this, we only have very scanty data

plotted.

In this study, NO surface concentrations did not show any statistically significant relationship with NO2- throughout the whole area, and similar variations of salinity, NO surface concentrations, and NO2-concentrations was found only from station B65 to station B70 where affected by the outflow of the Huanghe River in the southern BS (Figure 3). The photochemical production of NO was also affected by pH, temperature, irradiation density, concentration of CDOM, etc. besides NO2- concentration. Therefore, a comprehensive correlation might be difficult to obtain by in situ observation.

6. Page 2, Line 1: 10 % should be 10%. This should be applicable in other relevant places in the manuscript.

We will correct them as suggested. Thank you for your suggestion.

7. Page 3: we have interchangeable use of "h" and "hour(s)". The authors should stick to "h" preferably.

We decide to stick to "h" as suggested.

8. Page 6, Line 29: "0.00 × 10-11 molL-1s1" should be "not detectable"

We will correct it as suggested.

9. Page 7: The statement between Lines 8 and 9 should read ". . .but would also generate reactive oxygen species like O2âAËŸ c-, ROOâ′AËŸ c and ′ other OH-related radicals, which in turn, would efficiently scavenge NO. . ."

We will revise it as suggested. Thank you for your suggestion.

10. Page 7, Line 17: 24h should be 24 h 11. Page 10, Line 19: 13(4), 1-31 were repeated in the reference

We will correct them as suggested. Thank you for your comments and suggestions.

---

## Referee Comment (RC2) · Carolin Löscher (Referee) · 24 Sep 2019

Review of 'Nitric oxide (NO) in the Bohai and Yellow Seas' by Ye Tian et al

The manuscript by Ye Tian et al on NO distribution in the Bohai and Yellow Sea is a first report on the distribution of this intermediate of the nitrogen cycle, complemented with a model on production sources and pathways. The paper is well-written and straight forward to understand, it will certainly be interesting to the readers of Biogeosciences. I have some comments of rather technical nature, as well as some questions to the authors, which I hope to be perceived constructive for the quality of the manuscript.

Generally, I am wondering why there hasn't been any discussion of the NOx rates from anthropogenic sources as they were mentioned three times in the paper. This would possibly be important to do, particularly in the context of production pathways, which leads me to my second point. Most of the production seems to be photochemical in

surface waters. A possible change in dust particles or a change in UV intensity could both alter this production, I assume, to a more or less significant extent- could that be discussed? In addition, there is this mysterious pathway producing NO during dark periods. One option would be nitrification, which is in some steps light sensitive- this needs to be discussed from my perception. The obvious pathway, denitrification, does not seem to contribute anything here- why is that?

Technical considerations:

P1

L. 12 change 'pathways' to 'turnover'

L. 17 and throughout the text: There is a dot between mol and L, please remove

L. 21 and throughout the text, the commas are incorrect. Replace 'unknown' by 'unidentified'.

L. 23 The last sentence is repeated later in the manuscript and is not particularly informative as part of the abstract, I recommend removing it, here.

p. 2

L. 3 What would be the impact of NO in the atmosphere?

L. 14 'The current understanding...'

P. 3

L 13 ff. Please remove this list and show the stations on the map in Figure 1.

L. 22 Please explain what those chemicals are added for.

p. 4

l. 1 replace 'were' by 'was'

L. 3 the statement of the precision sounds weird, rephrase please.

p. 5

l. 16 This could still be explained, here and not only presented as a table, also the table content should be submitted to PANGAEA and a doi should go into the text, here.

Fig. 2 The panels could be smaller and shown side by side, also a colored figure would be beneficial, here.

L. 26 How would a correlation to depth make sense if we are talking about surface samples?

p. 7

L. 11 replace 'was' by 'were'

L. 13 Here, a discussion on the different biological pathways would fit in well.

L. 19 dot after ) is missing.

p. 8

L. 17/ 18 Awkward sentence, please rephrase

L. 21, same comma situation as in the abstract. Again, it's only unidentified, but possibly known. As no genetic or biological data is presented such a statement is not possible.

l. 24 This is the same sentence as in the abstract, the whole idea should be discussed, before, otherwise the statement is somewhat unfounded.

---

## Author Response (AR1)

Dear Prof. Wajih Naqvi,

We would like to thank you and the anonymous reviewer and Prof. Carolin Löscher for their comments and suggestions which helped us improve our manuscript. Please find our final responses (in red) to all comments (in black) in this document. The line numbers mentioned by the reviewers refer to the original version of the manuscript while the line numbers in our replies refer to the revised version of the manuscript.

**Response to reviewer #1.**

Comments from reviewer #1 are in black while our response in red and changes in the manuscript are in blue.

The reviewer found the manuscript acceptable since it provided relevant findings. However, the following observations are pointed out:

1.      The last sentence in the abstract seems spurious and unsubstantiated, because there was no part of the manuscript that reported research finds on $NO_x$ emissions from ships.

In our manuscript of Page 8 line 10-15, we compared the NO emission of the Yellow Sea and the Bohai Sea to the atmosphere based on our observation ($7.6 \times 10^7$ gN·month$^{-1}$) with the satellite-derived average $NO_x$ emission estimate (about $1.3 \times 10^{10}$ g N·month$^{-1}$) reported by Ding et al., 2018. Because Ding et al.'s (2018) estimate is dominated by the $NO_x$ emissions from ships' diesel engines, thus we concluded that the oceanic NO emissions to the atmosphere were negligible compared to anthropogenic $NO_x$ sources such as emissions from ships.

2.      Judging from the distribution patterns of NO flux and the wind speed (Figure 5), one would expect an important inference and conclusive statement in the abstract and discussion. Previous investigators (Anifowose AJ, Sakugawa H. 2017. Determination of Daytime Flux of Nitric Oxide Radical (NO·) at an Inland Sea-Atmospheric Boundary in Japan. J Aquat Pollut Toxicol., 1:2) reported wind speed as

an important factor governing NO flux at the air-sea interface.

Yes, wind speed was an important factor governing NO flux at the air-sea interface judging from the distribution patterns of NO flux and the wind speed. We have supplemented it in the revised manuscript and cited this reference of Anifowose and Sakugawa (2017).

The comparison of wind speeds and flux densities reveal that the flux densities are mainly driven by the wind speed (Fig. 7) and they showed significant positive relationship (SPSS v.16.0). Anifowose and Sakugawa (2017) also found that wind speed was an important factor governing NO flux at the air-sea interface.

3.      The analytical methods on the measurements of NO concentration and during irradiation experiments (photoformation rate) are not explicit enough. I understand that the authors referred to Liu et al. (2017), there is need for them to report detailed analytical procedures in the manuscript.

Thank you for your comment. We had supplemented the analytical methods on the measurements of NO concentration and during irradiation experiments in revised manuscript.

The photolysis experiments were conducted under natural light conditions on deck. Surface water samples were filtered with 0.45 μm Millipore membrane and transferred into 10 mL cleaned quartz vials wrapped with Al foil with no headspace. Then 200 μL NaN$_3$ solutions were added to remove the microbial influence. After adding 10 μL of $1\times10^{-3}$ mol L$^{-1}$ 2, 3-diaminonaphthalene (DAN, trap NO) solution and gentle mixing, the fluorescence of the mixed solution was measured before irradiation (Liu et al., 2017). Then the same sample without addition DAN of were capped with a Teflon-lined silicone septum and without Al foil, the vial was placed in a shallow circulating seawater bath. After 0.5 h irradiation time, 10 μL of $1\times10^{-3}$ mol L$^{-1}$ DAN were added and the fluorescence of the solution was analyzed. The NO concentrations were measured with the method described in Liu et al., (2017). The NO photolysis production rates were computed based on the time-dependent difference between the NO

concentrations before and after irradiation. For dark controls, vials were wrapped in Al-foil. The quartz vials and syringe used in the experiment were soaked in a 10% (v/v) HCl bath for 24 h, rinsed with Milli-Q water and baked at 500°C for 4 h.

4.      In view of comment 3 above, one would ask the precious question as to whether the measured concentration of NO during the irradiation experiment was steady-state concentration, even when the NO scavenging rate constant in the seawater (during the experiment) remained unknown?

During the irradiation of sunlight, production and consumption of NO occurs simultaneously in seawater bulk, for example, radicals from CDOM can scavenge NO, and thus the measured concentration of NO was a net value of NO production.

5.      I think there should be comprehensive correlation plot (and its discussion in the manuscript) of relationship between NO and $NO_2$(a major NO source). While it is true that the authors presented Figure 3 to reflect this, we only have very scanty data plotted. In this study, NO surface concentrations did not show any statistically significant relationship with $NO_2^-$ throughout the whole area, and similar variations of salinity, NO surface concentrations, and $NO_2^-$concentrations was found only from station B65 to station B70 where affected by the outflow of the Huanghe River in the southern BS (Figure 3). The photochemical production of NO was also affected by pH, temperature, irradiation density, concentration of CDOM, etc. besides $NO_2^-$ concentration. Therefore, a comprehensive correlation might be difficult to obtain by in situ observation.

6.      Page 2, Line 1: 10 % should be 10%. This should be applicable in other relevant places in the manuscript.

We have correct them as suggested. Thank you for your suggestion.

This reaction may account for 10% of nitrite loss in surface waters of the central equatorial Pacific (Zafiriou et al., 1980).

Throughout the manuscript, the space between number and % were removed.

7. Page 3: we have interchangeable use of "h" and "hour(s)". The authors should stick to "h" preferably.

We decide to stick to "h" as suggested.

The NO photolysis production rates were computed based on the time-dependent difference between the NO concentrations before and after irradiation. For dark controls, vials were wrapped in Al-foil. The quartz vials and syringe used in the experiment were soaked in a 10% (v/v) HCl bath for 24 h, rinsed with Milli-Q water and baked at 500°C for 4 h.

8. Page 6, Line 29: "$0.00 \times 10^{-11}$ mol L$^{-1}$s$^{-1}$" should be "not detectable"

We have corrected it as suggested.

The photoproduction rates of NO in the BS, the NYS, and the SYS varied from not detectable to $5.07 \times 10^{-11}$ mol L$^{-1}$ s$^{-1}$, 0.09 to $0.69 \times 10^{-11}$ mol L$^{-1}$ s$^{-1}$, and 0.32 to $1.54 \times 10^{-11}$ mol L$^{-1}$ s$^{-1}$, respectively.

9. Page 7: The statement between Lines 8 and 9 should read "…but would also generate reactive oxygen species like $O_2$•-, ROO• and other OH-related radicals, which in turn, would efficiently scavenge NO…"

We have revised it as suggested. Thank you for your suggestion.

The high illumination observed in the SYS (Table 2) does not only lead to enhanced NO photoproduction (see Table 2) but would also generate reactive oxygen species like $O_2$•-, ROO•, and other OH- related radicals, which in turn, would efficiently scavenge NO (Olasehinde et al, 2010)

10. Page 7, Line 17: 24h should be 24 h

We have corrected it as suggested. Thank you for your suggestions.

Together with the fact that that NO was supersaturated during daytime and nighttime during the 24 h station B65 this indicates that the BS and YS were a source of NO to the atmosphere.

11. Page 10, Line 19: 13(4), 1-31 were repeated in the reference

We have corrected it as suggested. Thank you for your comments and suggestions.

Liu, C.Y., Feng, W.H., Tian, Y., Yang, G.P., Li, P.F., and Bange, H.W.: Determination of dissolved nitric oxide in coastal waters of the Yellow Sea off Qingdao, Ocean Sci., 13(4), 1-31. https://doi.org/10.5194/os-2017-10, 2017.

**Response to Prof. Carolin Löscher.**

Comments from reviewer #1 are in black while our response in red and changes in the manuscript are in blue.

The manuscript by Ye Tian et al on NO distribution in the Bohai and Yellow Sea is a first report on the distribution of this intermediate of the nitrogen cycle, complemented with a model on production sources and pathways. The paper is well-written and straight forward to understand, it will certainly be interesting to the readers of Biogeosciences.

Thank you very much for your comments and suggestions. The manuscript was amended, and you will find a detailed description in how we took all the comments and suggestions into account in the preparation of the revised manuscript.

I have some comments of rather technical nature, as well as some questions to the authors, which I hope to be perceived constructive for the quality of the manuscript. Generally, I am wondering why there hasn't been any discussion of the NOx rates from anthropogenic sources as they were mentioned three times in the paper. This would possibly be important to do, particularly in the context of production pathways, which leads me to my second point.

In our manuscript 3.4 "sea-to-air fluxed of NO" part, we describe the anthropogenic source from the ship engine in the Bohai and Yellow Seas as "Ding et al. (2018) report a satellite-derived average $NO_x$ emission estimate for June over the BS and the YS in the period from 2007 to 2016 of about $1.3 \times 10^{10}$ g N month$^{-1}$. The obvious very large discrepancy between the satellite-derived emission estimate and the one presented here results from the fact that Ding et al.'s (2018) estimate is dominated by the $NO_x$ emissions from ships' diesel engines. This indicates that oceanic NO emissions to the atmosphere only account for a negligible fraction (~0.6%) of the $NO_x$ emissions observed over the BS and YS."

Most of the production seems to be photochemical in surface waters. A possible change in dust particles or a change in UV intensity could both alter this production, I assume, to a more or less significant extent- could that be discussed?

We have added the discussion about dust particles, however, we did not do the detailed researches about this, thus we cited Olasehinde et al. (2010) and Liu et al. (2017) to elaborate the influence of the dust particles.

Besides, Olasehinde et al. (2010) found that filtered and unfiltered seawater samples collected from the Seto Inland Sea showed no significant difference in NO• photoformation rates, which suggested a negligible contribution of NO• produced by photobiological processes from particle matter in seawater. However, Liu et al. (2017) reported that the rates difference between filtered (0.45 μm, $1.46\times10^{-12}$ mol $L^{-1}$ $s^{-1}$) and unfiltered ($1.52\times10^{-12}$ mol $L^{-1}$ $s^{-1}$) seawater samples from coastal waters of the Yellow Sea indicated that particles in seawater could increase the NO production rate. The difference might be due to the composition of sample, filter membrane, etc. Thus, further research is needed.

We used the UV data from ECMWF reanalysis data sets (ERA−5 hourly mean surface downward UV radiation flux data) and analyzed the photoproduction rates with UV data. The NO photoproduction rates showed a positive relationship with the mean surface downward UV radiation flux ($p < 0.01$, $r = 0.865$, $n = 26$),

Overall, the NO photoproduction rates showed a positive relationship with illuminance ($p < 0.01$, $r = 0.884$, $n = 26$) and the mean surface downward UV radiation flux ($p < 0.01$, $r = 0.865$, $n = 26$) indicating that the NO concentrations in the surface layer during daylight were dominated by photochemical production.

The average photoproduction rate in the NYS was obviously lower than those in the BS and the SYS, consistent with the average illuminances of 22450, 20433, and 27825 lx and the mean surface downward UV radiation flux 34.7, 32.1, and 40.6 W $m^{-2}$ for the BS, the NYS and the SYS, respectively.

In addition, there is this mysterious pathway producing NO during dark periods. One option would be nitrification, which is in some steps light sensitive- this needs to be discussed from my perception.

We have added some discussion like nitrification process and some chemical process into our manuscript.

This suggests that there was also a non-photochemical NO dark production in the surface layer like nitrification process or other chemical processes like the process of ammonium ($NH_4^+$/$NH_3$) oxidation into $NO_2^-$ and $NO_3^-$ (Joussotdubien and Kadiri, 1970). Caranto and Lancaster (2017) found that NO was an obligate bacterial nitrification intermediate produced by hydroxylamine oxidoreductase. Ward and Zafiriou (1988) also found that NO might play as an intermediate of a soluble byproduct of nitrification such as hydroxylamine.

The obvious pathway, denitrification, does not seem to contribute anything here- why is that?

In our study area, DO fluctuated from 178 to 271 μmol $L^{-1}$ in the surface water and 147 to 266 μmol $L^{-1}$ in the bottom water (added into the manuscript), which indicated that the water was well-oxygenated thus it seemed that denitrification could not occur. In our manuscript, "The water columns of the BS and YS were well-oxygenated during our study and, thus, no suboxic or anoxic conditions were detected. Therefore, we did not found any enhancement of NO concentrations in the bottom layers with low DO or $NO_2^-$ concentrations as observed in the OMZ of the eastern tropical South Pacific Ocean off Peru (Lutterbeck et al., 2018)", which partly explained this.

Technical considerations:

P1

L. 12 change 'pathways' to 'turnover'

We have corrected it and thank you.

however, our knowledge about its oceanic distribution and turnover is rudimentary.

L. 17 and throughout the text: There is a dot between mol and L, please remove

Thank you, we have removed the dot between mol and L throughout the text.

L. 21 and throughout the text, the commas are incorrect. Replace 'unknown' by 'unidentified'.

We have corrected it and thank you.

The persistent nighttime NO supersaturation pointed to an unidentified NO dark production.

L. 23 The last sentence is repeated later in the manuscript and is not particularly informative as part of the abstract, I recommend removing it, here.

We have removed it as you advised.

p. 2

L. 3 What would be the impact of NO in the atmosphere?

We have added the NO environmental influence to the revised ms.

Zafiriou and McFarland (1981) suggested that photochemically produced NO is a potential source of atmospheric NO during daylight, which could further lead to ozone hole, acid precipitation and photochemical smog (Bange, 2008).

L. 14 'The current understanding…'

We have revised it and thank you.

The current understanding of the oceanic NO distribution is mainly limited to the ocean surface (Zafiriou and McFarland, 1981; Olasehinde at al., 2009; 2010; Liu et al., 2017) and oxygen minimum zones (Ward and Zafiriou, 1988; Lutterbeck et al., 2018).

P. 3

L 13 ff. Please remove this list and show the stations on the map in Figure 1.

We have corrected it and thank you.

[Figure]

**Figure 1.** Locations of the sampling stations in the BS and the YS during summer. Solid dots (●) represent the stations for incubation experiments.

L. 22 Please explain what those chemicals are added for.

The NaN₃ solutions were added to remove the microbial influence while DAN were added to react with NO (process of trapping NO) and the fluorescence of the product was measured using a method described by Liu et al., 2017.

Then 200 μL NaN₃ solutions were added to remove the microbial influence. After adding 10 μL of 1×10⁻³ mol L⁻¹ 2, 3-diaminonaphthalene (DAN, trap NO) solution and gentle mixing, the fluorescence of the mixed solution was measured before irradiation (Liu et al., 2017).

p. 4

L. 1 replace 'were' by 'was'

We have corrected it and thank you.

The concentrations of dissolved inorganic nitrogen (nitrate, nitrite, and ammonium) was analyzed using a nutrient automatic analyzer (Auto Analyzer 3, SEAL Analytical, USA) in the laboratory.

L. 3 the statement of the precision sounds weird, rephrase please.

We have revised "higher" to "better than".

The detection limits were 0.14 μmol $L^{-1}$ for nitrate, nitrite, and ammonium, with the precision of the method better than 3% (Liu et al., 2005).

p. 5

L. 16 This could still be explained, here and not only presented as a table, also the table content should be submitted to PANGAEA and a doi should go into the text, here.

We have added some information and explanation here and the data was submitted to Pangaea and we are still waiting for the doi (https://issues.pangaea.de/browse/PDI-21749).

In the study area, temperature varied from 8.8 to 21.7°C and salinity varied from 29.57 to 32.73‰ in the surface water. DO fluctuated from 178 to 271 μmol $L^{-1}$. The average concentrations of $NH_4^+$-N, $NO_2^-$-N, and $NO_3^-$-N were 2.11, 0.20, and 2.59 μmol $L^{-1}$, respectively. While temperature varied from 3.8 to 21.6°C, salinity varied from 30.00 to 33.71‰, and DO varied from 147 to 266 μmol $L^{-1}$ in the bottom water.

Fig. 2 The panels could be smaller and shown side by side, also a colored figure would be beneficial, here.

We have redrawn the colored figure and made them side by side.

[Figure]

**Figure 2.** Horizontal distributions of NO (pmol L⁻¹) in the surface and bottom layers.

L. 26 How would a correlation to depth make sense if we are talking about surface samples?

We have removed the description of the correlation between the depth and the NO concentration in the surface water.

p. 7

L. 11 replace 'was' by 'were'

We have corrected 'was' into 'were' and thank you.

Overall, the NO photoproduction rates showed a positive relationship with illuminance ($p < 0.01$, $r = 0.884$, $n = 26$) and the mean surface downward UV radiation flux ($p < 0.01$, $r = 0.884$, $n = 26$) indicating that the NO concentrations in the surface layer during daylight were dominated by photochemical production.

L. 13 Here, a discussion on the different biological pathways would fit in well.

Thank you for your suggestion. We have revised this part about the nitrification process here.

This suggests that there was also a non-photochemical NO dark production in the surface layer like nitrification process or other chemical processes like the process of

ammonium ($NH_4^+$/$NH_3$) oxidation into $NO_2^-$ and $NO_3^-$ (Joussotdubien and Kadiri, 1970). Caranto and Lancaster (2017) found that NO was an obligate bacterial nitrification intermediate produced by hydroxylamine oxidoreductase, Ward and Zafiriou (1988) also found that NO might play as an intermediate of a soluble byproduct of nitrification such as hydroxylamine.

L. 19 dot after ) is missing.

We have corrected it and thank you.

p. 8

L. 17/18 Awkward sentence, please rephrase

We rephrased it as "The horizontal distribution of NO concentration in both the surface and bottom layers were highly variable, however, there was no significant difference between the mean NO concentrations in the surface and bottom layers."

L. 21, same comma situation as in the abstract. Again, it's only unidentified, but possibly known. As no genetic or biological data is presented such a statement is not possible.

We have corrected "unknown" into "unidentified" and the comma situation has also been revised.

The persistent nighttime NO supersaturation pointed to a non-photochemical (so far unidentified chemical and/or biological) NO dark production.

L. 24 This is the same sentence as in the abstract, the whole idea should be discussed, before, otherwise the statement is somewhat unfounded.

As mentioned above, $NO_x$ emissions from ship's diesel engines were described as "
[revised manuscript text omitted]

---

## Author Response (AR2)

Dear Prof. Wajih Naqvi,

We would like to thank you and the anonymous reviewer for the suggestions which helped us improve our manuscript. Please find our responses (in red) to the comments (in black) from the reviewer and our manuscript changes (in blue) in this document. The line numbers mentioned by the reviewers refer to the original version of the manuscript while the line numbers in our replies refer to the revised version of the manuscript.

**Response to reviewer #1.**

1. In the abstract, the authors used detection limit (Line 15), but consistently used limit of detection (LOD) in the body of the manuscript. I suggest use of LOD too in the abstract for the sake of uniformity.

Thank you for your advice, we have revised it and used limit of detection in the abstract.

The NO concentrations in the surface and bottom layers were highly variable and ranged from below the limit of detection (i.e. 32 pmol L$^{-1}$) to 616 pmol L$^{-1}$ in the surface layer and to 482 pmol L$^{-1}$ in the bottom layer. There was no significant difference ($p > 0.05$) between the mean NO concentrations in the surface (186 ± 108 pmol L$^{-1}$) and bottom (174 ± 123 pmol L$^{-1}$) layers.

2. Page 1, Line 16 and page 5, Line 30 and Page 6, Line 1: the confidence level (p values) of the quoted statistically significant differences/relationship should be provided.

Thank you and we have added the confidence level into the revised manuscript.

The NO concentrations in the surface and bottom layers were highly variable and ranged from below the limit of detection (i.e. 32 pmol L$^{-1}$) to 616 pmol L$^{-1}$ in the surface layer and to 482 pmol L$^{-1}$ in the bottom layer. There was no significant difference ($p > 0.05$) between the mean NO concentrations in the surface (186 ± 108 pmol L$^{-1}$) and bottom (174 ± 123 pmol L$^{-1}$) layers.

Overall, there were no statistically significant differences ($p > 0.05$) between the mean NO concentrations in the surface and bottom layers.

NO surface concentrations did not show any statistically significant relationship ($p > 0.05$) with sea surface temperature, salinity, DO, Chl-*a*, and illuminance.

3. Page 3: Authors should include (under section 2.2) short sentence(s) describing preservation/storage of the seawater ahead of the laboratory measurements of $NH_4^+$-N, $NO_2^-$-N and $NO_3^-$-N.

We have included the preservation method of the $NH_4^+$-N, $NO_2^-$-N, and $NO_3^-$-N.

Samples for the analysis of $NH_4^+$-N, $NO_2^-$-N, and $NO_3^-$-N were immediately filtered through filters (pore size 0.45 μm, pre-ignited at 450 °C for 6 h). The filtrates were stored in polyethylene bottles (pre-marinated with 1:10 HCl for 24 h) at − 20 °C.

4. Page 3, Line 22: "trap NO" should be "a NO-probe"

Thank you and we have replaced "trap NO" with "a NO-probe".

After adding 10 μL of $1×10^{-3}$ mol $L^{-1}$ 2, 3-diaminonaphthalene (DAN, a NO-probe) solution and gentle mixing, the fluorescence of the mixed solution was measured before irradiation (Liu et al., 2017).

5. Page 3, Line 24: "After 0.5 h irradiation time, …" which of the samples were the author referring to?

We referred to the sample "the duplicate sample without addition of DAN were capped with a Teflon-lined silicone septum and without Al foil, and the vial was placed in a shallow circulating seawater bath", which were added the DAN after the irradiation. This fluorescence value was applied to calculate the apparent production amount of NO, then the difference between the NO concentrations before and after irradiation was calculated and the apparent production rates were calculated based on this.

After 0.5 h irradiation time, 10 μL of $1×10^{-3}$ mol $L^{-1}$ DAN were added into the duplicate sample without addition of DAN before and the fluorescence of the solution was analyzed.

6. Page 6, Line 7: "we did not found" should be "we did not find".

We have revised it and thank you.

[revised manuscript text omitted]